# Dissecting Tumor Heterogeneity by Liquid Biopsy—A Comparative Analysis of Post-Mortem Tissue and Pre-Mortem Liquid Biopsies in Solid Neoplasias

**DOI:** 10.3390/ijms26157614

**Published:** 2025-08-06

**Authors:** Tatiana Mögele, Kathrin Hildebrand, Aziz Sultan, Sebastian Sommer, Lukas Rentschler, Maria Kling, Irmengard Sax, Matthias Schlesner, Bruno Märkl, Martin Trepel, Maximilian Schmutz, Rainer Claus

**Affiliations:** 1Pathology, Faculty of Medicine, University of Augsburg, Stenglinstr. 2, 86156 Augsburg, Germany; tatiana.moegele@uk-augsburg.de (T.M.); kathrin.hildebrand@uk-augsburg.de (K.H.); lukas.rentschler@uk-augsburg.de (L.R.); maria.eberle@uk-augsburg.de (M.K.); bruno.maerkl@uk-augsburg.de (B.M.); 2Bavarian Cancer Research Center (BZKF), 91054 Erlangen, Germany; martin.trepel@uk-augsburg.de (M.T.); maximilian.schmutz@uk-augsburg.de (M.S.); 3Hematology and Oncology, Faculty of Medicine, University of Augsburg, 86156 Augsburg, Germany; aziz.sultan@uk-augsburg.de; 4MVZ-Onkologie im Elisenhof, 80335 Munich, Germany; s.sommer@onkologie-elisenhof.de; 5Biomedical Informatics, Data Mining and Data Analytics, University of Augsburg, 86159 Augsburg, Germany; irmengard.sax@uni-a.de (I.S.); matthias.schlesner@uni-a.de (M.S.); 6Comprehensive Cancer Center Augsburg (CCCA), 86156 Augsburg, Germany

**Keywords:** tumor heterogeneity, clonal evolution, liquid biopsy, NGS, mutational landscape and molecular pathology

## Abstract

Tumor heterogeneity encompasses genetic, epigenetic, and phenotypic diversity, impacting treatment response and resistance. Spatial heterogeneity occurs both inter- and intra-lesionally, while temporal heterogeneity results from clonal evolution. High-throughput technologies like next-generation sequencing (NGS) enhance tumor characterization, but conventional biopsies still do not adequately capture genetic heterogeneity. Liquid biopsy (LBx), analyzing circulating tumor DNA (ctDNA), provides a minimally invasive alternative, offering real-time tumor evolution insights and identifying resistance mutations overlooked by tissue biopsies. This study evaluates the capability of LBx to capture tumor heterogeneity by comparing genetic profiles from multiple metastatic lesions and LBx samples. Eight patients from the Augsburger Longitudinal Plasma Study with various types of cancer provided 56 postmortem tissue samples, which were compared against pre-mortem LBx-derived circulating-free DNA sequenced by NGS. Tissue analyses revealed significant mutational diversity (4–12 mutations per patient, VAFs: 1.5–71.4%), with distinct intra- and inter-lesional heterogeneity. LBx identified 51 variants (4–17 per patient, VAFs: 0.2–31.1%), which overlapped with mutations from the tissue samples by 33–92%. Notably, 22 tissue variants were absent in LBx, whereas 18 LBx-exclusive variants were detected (VAFs: 0.2–2.8%). LBx effectively captures tumor heterogeneity, but should be used in conjunction with tissue biopsies for comprehensive genetic profiling.

## 1. Introduction

Tumor heterogeneity refers to the diverse genetic, epigenetic, and phenotypic variations exhibited by malignant cell populations [1,2]. This phenomenon manifests at multiple levels: inter-tumor heterogeneity describes variability among patients with the same histologically defined cancer type, whereas intra-tumor heterogeneity refers to differences among tumor cells within an individual patient [3,4]. Spatially, these variations can be detected between metastatic sites (inter-lesional) and within a single lesion (intra-lesional) [5,6]. Moreover, tumors evolve dynamically over time under selective pressures such as therapy and changes in the tumor microenvironment, resulting in temporal heterogeneity referred to as clonal evolution [7,8].

Clinically, tumor heterogeneity has significant implications for treatment response and the emergence of resistance. Molecular differences within the tumor can lead to “mixed” responses, where some lesions respond while others progress [9,10,11]. Resistance often arises from newly emerging subclones that evade targeted therapies, ultimately diminishing treatment efficacy [12,13].

Advances in high-throughput technologies such as next-generation sequencing (NGS) have facilitated comprehensive molecular characterizations of tumors [14]. However, routine use of repeated or multiple tissue biopsies—required to capture both spatial and temporal heterogeneity—remains clinically challenging due to invasiveness and logistical constraints [15,16]. As a result, a single tissue biopsy (TBx) often provides only a limited snapshot of a tumor’s complete molecular landscape.

By contrast, liquid biopsy (LBx) offers a minimally invasive approach to assess the comprehensive genetic profile of solid tumors. Circulating tumor DNA (ctDNA), released through processes such as apoptosis and necrosis, can be readily detected in the bloodstream, providing real-time insights into the evolving tumor genome [17,18]. Studies have demonstrated the potential of LBx to identify clinically relevant alterations, including emerging resistance mutations that may be missed by a single tissue biopsy [19,20]. For instance, one report demonstrated a decline in a known resistance mutation (mutant *MEK1*) under treatment while a previously undetected *KRAS* mutation emerged in a non-responding metastasis [21]. In patients with gastrointestinal cancers who develop acquired resistance to targeted therapies, liquid biopsy (LBx) detected resistance mutations absent in matched tissue biopsies (TBx) in up to 78% of cases [22]. Building on this, we investigated the efficacy of LBx in capturing inter- and intra-tumor heterogeneity by comparing genetic profiles across multiple tumor lesions within individual patients representing diverse cancer types. Specifically, we contrasted the sum of clonal alterations detected across tissue samples with the mutational landscape observed in LBx. Our goal was to determine whether LBx can sufficiently represent, resolve, and deconstruct the spatial heterogeneity of solid tumors for routine clinical decision-making.

## 2. Results

### 2.1. Patient Characteristics and Sampling

The study cohort comprised seven patients with various tumor entities. Mean age at diagnosis was 56.4 years, ranging from 41 to 76 years. All participants were Caucasian, with one female patient.

Lung cancer was the most common tumor entity, accounting for 43% of cases (*n* = 3), including two patients with lung adenocarcinoma (LUAD; Patients 2 and 4) and one with lung squamous cell carcinoma (LUSC; Patient 5). Two patients had gastrointestinal tumors: colorectal adenocarcinoma (COADREAD; Patient 1) and pancreatic adenocarcinoma (PAAD; Patient 7). One patient was diagnosed with follicular thyroid carcinoma (THFO; Patient 6), and another patient (Patient 3) had an adenocarcinoma of unknown primary (ADNOS).

Survival times, measured from diagnosis to death, ranged from 23 days to 186 months (mean: 1044 days). Patient 6 (THFO) had the longest survival, while Patient 4 (ADNOS) experienced the shortest survival, passing away shortly after diagnosis due to rapid disease progression. Detailed individual treatment timelines are provided in Appendix A.

All patients were diagnosed with stage IV disease at enrollment, presenting with distant metastases. A total of 56 postmortem biopsy samples were collected, predominantly from the lungs (25%), liver (17.9%), and lymph nodes (10.7%). Additional biopsies were obtained from less common sites, including the spleen, epicardium, prostate, and peritoneum. Sampling focused on biopsy sites showing radiological evidence of disease progression, but additional samples from stable or regressing lesions, as well as previously undetected metastases, were also included. Intra-lesional biopsies aimed at assessing within-lesion heterogeneity were performed in five patients, primarily targeting progressive lesions. Further details on biopsy locations, radiological findings, and intra-lesional sampling are provided in Appendix A. Additionally, pre-mortem liquid biopsies (blood samples) were collected from each patient between 6 and 74 days prior to death (mean: 27 days).

### 2.2. Metastatic Lesions Demonstrate Heterogeneity to Variable Degrees

We first characterized the mutational landscape and intra-tumoral heterogeneity based on TBx. All detected mutations, alterations, and their respective frequencies are summarized in Appendix A. The mean read depth (DP) was 1873 (range: 336–5389). *TP53* variants were most frequently observed and were present in nearly all lesions within individual patients. Approximately 50% of identified mutations were consistently present across all investigated lesions per patient, including both missense and synonymous alterations. Notably, in six cases, a specific mutation was absent in a singular lesion, whereas 15 mutations were found exclusively in individual biopsy samples. This highlights considerable intra-tumoral heterogeneity.

We further assessed the VAFs within each patient’s biopsies to understand intra-tumoral heterogeneity in greater detail. Hierarchical clustering identified distinct mutational profiles among samples, reflecting variability both within individual patients (intra-patient heterogeneity) and between different lesions (inter-lesional heterogeneity) (Figure 1; Appendix A).

We highlight patients 4 and 5 as representative cases. Biopsies of Patient 4 formed two distinct clusters (Figure 1A). One cluster, characterized by uniformly low VAFs (0–10%), included mediastinal lymph nodes and the right adrenal gland. The second cluster, with notably higher VAFs, particularly for *CPXCR1* c.323 A>T (ranging from 35.1% to 58.1%) and *TP53* c.413 C>T (ranging from 39.8% to 50.4%) mutations, predominantly encompassed left-sided lesions (lung, adrenal gland, rib) and both liver metastases. Interestingly, despite radiologically stable appearances, the mutational profiles of the two adrenal glands differed substantially, indicating pronounced clonal evolution.

Patient 5 demonstrated substantial variation in VAFs (ranging from 5.7% for *ZNF521* c.2401 C>T to 71.4% for *TP53* c.832 C>G) (Figure 1B). The prostate lesion distinctly clustered apart, exhibiting highly prevalent *APC* c.7504 G>A (45.5%) and *BRCA2* c.8851 G>A mutations (45.5% and 59.2%, respectively). The left kidney harbored both mutations with high VAF (e.g., *FBXW7* c.1556 A>G (68.8%) and *TP53* c.832 C>G (71.4%)) and alterations exclusive to that site with lower VAF (e.g., *GRM8* c.2315 G>C (26.1%)). Lung lesions showed variable VAFs but clustered closely with the right kidney lesion.

Patients 1 and 2 shared patterns akin to Patient 5, with unique VAF shifts in specific anatomical sites, such as the mesenterial lymph node in Patient 1 and the spleen in Patient 2. Patient 3 displayed consistently low VAFs, primarily for *TP53* c.743 G>A (14.2–19.2%). Meanwhile, Patients 6 and 7 formed distinct clusters, driven by dominant mutations in *KEAP1* c.781 C>T, *TP53* c.743 G>A, and *KRAS* c.34 G>C, respectively (Appendix A). These patterns emphasize the variable degrees of genomic divergence among metastatic lesions.

### 2.3. Liquid and Tissue Biopsies Reveal Partially Overlapping Mutation Profiles

Next, we compared genetic alterations identified in LBx relative to those in TBx (Figure 2). The average DP of LBx was 5589 (range: 1774–11,081) and detection sensitivity < 0.1%. The numbr of mutations exclusively found in LBx varied notably across patients, ranging from zero (Patient 6) to six (Patient 4), with a total of 18 somatic variants (mean VAF = 0.05%, range: 0.2–2.8%) (Appendix A). Among the variants identified, one, located in *KIT*, overlapped with genes associated with clonal hematopoesis of indeterminate potential (CHIP) and can therefore not be confidently associated as being of tumor origin. The number of mutations uniquely detected in TBx ranged similarly, from one (Patient 6) to eight (Patient 5) (Appendix A). Notably, the VAFs of mutations exclusively detected in TBx (mean VAF: 15.4%) were significantly lower compared to all TBx (mean VAF: 24.5%) detected alterations (*p* < 1 × 10^−6^). The number of overlapping mutations detected both in LBx and TBx (cross-section (CS)) ranged from two (Patient 3) to 11 (Patient 4). On average, 67% of all LBx-detected variants (range: 50–100%) were also found in TBx. In contrast, 64.5% of TBx-detected variants (range: 20–92%) overlapped with LBx findings. These results highlight a high degree of concordance between TBx and LBx while underscoring their complementary nature in capturing the full spectrum of tumor heterogeneity.

### 2.4. LBx and Its Potential to Mitigate the Risk of Missing Clinically Relevant Mutations in Single-Lesion Tissue Biopsies

Understanding inter-lesional heterogeneity is critical for accurate molecular profiling and was clearly evident in our cohort. Notably, more than 27% of the mutations detected by both LBx and TBx were not consistently found across all lesions within individual patients. Patients 2 and 6 presented uniform mutation profiles across all analyzed lesions, indicating lower heterogeneity. In contrast, other patients showed significant variability. For instance, in Patient 1, biopsy of the lymph node alone (TBx6) would have missed three mutations; notably, *POM121L* c.849 C>T was uniquely detected in one TBx and the LBx. Similarly, in Patient 5, the *TP53* c.832 C>G variant was only identified in the left kidney biopsy and by LBx. Patient 3 exhibited partial lesion-specificity for the *KIT* c.2278 G>A mutation, found in only half of the biopsies, and in Patient 7, the *RET* c.1701 C>T mutation was absent in biopsies from primary lesions. In Patient 4, detection varied substantially across different lesions (Figure 3). These findings underscore the importance of multi-lesion sampling to adequately capture tumor heterogeneity. Relying on a single TBx may overlook critically relevant mutations due to substantial inter-lesional variability and highlights the role of LBx in helping to overcome the limitations of localized tissue sampling.

### 2.5. Correlation Between LBx and TBx Partly Reflects the Composition of the Tumor

Considering both the intra-lesional heterogeneity and the variability in DNA shedding and degradation across tumor sites, we questioned the extent to which LBx reflects the overall tumor composition. Therefore, we evaluated the correlation of VAFs between TBx and LBx mutations shared by both methods using regression analysis (Figure 4A). Patient 4 showed strong correlations with R^2^ values ranging from 0.680 to 0.966. Conversely, Patient 5 exhibited consistently weak correlations across all samples. Patients 1, 2, and 6 demonstrated variable correlation strengths, with sporadic R^2^ values above 0.5. Correlation analysis was limited for Patient 3 due to few shared mutations. Patient 7, despite limited available data points, exhibited notably high correlations (R^2^ = 0.885–0.999). A full summary of these results is presented in Figure 4B. These findings emphasize patient-specific variability in the concordance between TBx and LBx, suggesting that, although not reliable across all patients, LBx can reflect the overall tumor composition, overcoming the limitations of single-lesion biopsy.

## 3. Discussion

Overcoming spatial tumor heterogeneity through genetic profiling via LBx presents a promising approach to understanding the complex and dynamic mutational landscape of solid tumors [23,24]. With our comparative analysis of post-mortem TBx and pre-mortem LBx, we aimed to evaluate the potential of LBx to overcome the limitations of spatial sampling and better reflect tumor-wide heterogeneity. LBx has the potential to estimate tumor evolution, therapeutic response, and the molecular variability that underpins resistance mechanisms missed by single TBx, which remains the clinical standard.

Tumor evolution is shaped by prolonged disease progression and multiple lines of therapy, both of which are expected to contribute to diverse genetic profiles across metastatic sites [25,26,27]. The long period of tumor development, coupled with multiple therapeutic interventions, contributes to a shifting mutational landscape and the appearance of novel subclones [28,29]. Here, we included both patients with relatively short and longer treatment periods with several lines of therapies. Although our cohort showed little evidence of therapy-specific molecular evolution in the form of resistance mechanisms, therapy-induced heterogeneity has been well documented in previous studies [30,31]. Other factors, such as the tumor microenvironment or intrinsic genomic instability, also play a role in shaping clonal dynamics [32,33]. Our data clearly confirms that genetic alterations are not uniformly distributed across lesions, with notable variations in both the number and type of mutations across intra- and inter-lesional samples. This becomes exceptionally evident in the case of the patient with a history of prostate cancer (Patient 5) presumed to have been curatively treated by definitive radiotherapy. A prostate lesion exhibited a completely distinct molecular profile not observed in other biopsied lesions, underscoring the risk of underestimating heterogeneity if relying on a single biopsy. Therefore, despite the advantages of TBx in providing spatial context to tumor mutations, its feasibility in routine clinical practice is constrained by invasive sampling procedures and accessibility limitations [34,35]. In contrast, LBx offers a non-invasive and readily available alternative that effectively captures a broader mutational landscape of tumors. This approach can yield deeper insights in multi-site disease settings, especially in a scenario with an unexpected tumor site, thereby addressing the limitations of conventional tissue methods [36,37]. High concordance rates for genomic alterations detected in LBx and TBx underpin its great potential [38].

Here, our findings reveal a complex interplay between TBx heterogeneity and LBx as described in the literature previously [38,39,40,41,42]. Overall, heterogeneity observed in TBx was also reflected in LBx. Notably, over 38% of mutations detected in common (LBx and TBx) were missing in at least one of the eight lesions analyzed. In two cases, a *KIT* c.2278 G>A mutation in Patient 1 and a *TP53* c.832 C>G mutation in Patient 5, the alterations were solely detectable in a single localization as well as by LBx. This highlights the risk of missing clinically relevant mutations when relying on a single lesion biopsy and, moreover, underscores the risk of underestimating tumor heterogeneity. 

Importantly, the mutational profile observed in LBx showed a positive correlation with the heterogeneity seen in solid tumor lesions, as demonstrated by regression analyses. However, this relationship was not consistent across all cases. For example, in Patient 5, despite having multiple TBx samples from different tumor areas, the VAFs did not correlate with those found in LBx. In contrast, Patient 4 displayed a strong correlation between TBx and LBx VAFs, even though tissue samples were taken from five different organs. This finding is particularly notable.

Several biological factors may explain these differences. Tumor size and growth rate affect the amount of cell death, which is the main source of ctDNA released into the bloodstream [43,44]. Supporting this, mutations detected only in tissue biopsies had significantly lower VAFs, suggesting that mutations with low VAF in tissue may be missed in LBx. However, some mutations were also found esxclusively in LBx. Additionally, tumors with high microsatellite instability tend to release more ctDNA, causing variability in ctDNA shedding among different cancer types [45]. Another important factor is tumor vascularization and proximity to blood vessels, as efficient blood flow can facilitate the release of ctDNA into circulation.

This can impact measurements even at a lesion-specific level [46] as exemplified by the clinically relevant absence of the *BRCA2* c.8851 G>A mutation in the LBx of Patient 5. Moreover, several studies have investigated the relationship between ctDNA levels and radiological assessments of disease stability or progression, generally finding that patients with progressive disease show elevated ctDNA levels compared to those with stable disease [47]. However, in our cohort, no consistent association was observed between radiological findings and the correlation results between LBx and TBx data (Appendix A). 

Numerous mutations were exclusively detected in TBx and missed by LBx (Appendix A). However, these alterations exhibited significantly lower VAFs, which may indicate their origin in subclonal populations. Given their low VAFs, the clinical significance of these mutations for therapeutic decision-making remains uncertain.

Another key observation of our study is the identification of unique aberrations in the LBx. The presence of LBx-exclusive mutations can result from the inability to sample all tumor mass from all manifestations. This assumption is supported by the patient-specific observations in Patient 5. Here, extensive tissue sampling was performed, and only a few unique LBx variants were identified, suggesting that LBx helps compensate for lesions in less-accessible or unrecognized metastases. However, detection of alterations in LBx with low VAFs must be critically questioned concerning biological relevance in the context of analytical hypersensitivity. On one hand, highly sensitive techniques such as NGS enable the detection of minute DNA fragments; among them, CAPP-seq (Cancer Personalized Profiling by deep Sequencing) offers enhanced sensitivity and specificity for identifying low-frequency mutations in ctDNA [48]. On the other hand, the clinical relevance of such low-VAF findings is not always clear [49,50]. 

Also, discriminating mutations related to CHIP by paired plasma and peripheral blood cell sequencing is crucial to avoid misinterpretation of results [49,51]. In response to the potential confounding role of CHIP in interpreting LBx-exclusive mutations, we systematically evaluated all alterations listed in Appendix A against a curated set of genes frequently associated with CHIP. This list includes canonical drivers such as *DNMT3A*, *TET2*, *ASXL1*, *JAK2*, *TP53*, *SF3B1*, *PPM1D*, *SRSF2*, *IDH1*, *IDH2*, and *U2AF1*, among others, commonly implicated in age-related clonal expansions of hematopoietic lineages. Among the variants identified, only one—located in *KIT*—overlapped with this CHIP-associated gene set. However, this variant was classified as a variant of unknown significance (VUS) and thus is unlikely to have clinical relevance in the present context, particularly regarding drug resistance. Based on current evidence, we consider it unlikely that CHIP-associated mutations significantly influenced our LBx findings. Nonetheless, we emphasize the importance of incorporating paired plasma and leukocyte sequencing in future studies to more definitively distinguish tumor-derived from hematopoietic-origin variants. Furthermore, there is a risk of overinterpreting noise as a meaningful signal, so, considering orthogonal assays for a second opinion to strengthen confidence in low-VAF results is crucial before making premature clinical decisions.

There are some limitations to our findings. Firstly, the analysis was based on a small cohort of patients with varying tumor types. Although we included a large number of biopsies from various sites to obtain a comprehensive overview of each patient, the entire landscape of the tumor’s manifestation may not have been fully captured, as not all lesions were sampled. Notably, the patient with a history of presumed curatively treated prostate cancer exemplifies significant intra-patient heterogeneity, which may remain undetected by conventional TBx but could potentially be revealed through LBx.

Our study offered a unique opportunity of sampling patients from the Augsburg Longitudinal Plasma Study (ALPS) cohort by full autopsy. Its primary strength lies in its technical design. By including multiple tumor entities, examining samples from multiple tumor lesions (including an intralesional sample), and harmonizing the biochemical workflow with identical library and panel technology for both, TBx and LBx, we ensured robust comparability. In addition, all variants identified in the raw data were manually validated to ensure accuracy.

## 4. Methods and Materials

### 4.1. Ethics Approval and Consent to Participate

All patients included in this study are participants in the Augsburg Longitudinal Plasma Study (ALPS) [52]. All participants have provided written informed consent for the ALPS trial and the Augsburg Central Biobank (ACBB). This trial adheres to the principles of the Declaration of Helsinki and was approved by the Local Ethics Committee (Ethikkommission der Ludwig-Maximilians-Universität München, (approval no. 20-0972)) and registered at clinicaltrials.gov (ClinicalTrials.gov Identifier: NCT05245136, registered on 8 November 2021). The selected patients provided additional consent for post-mortem autopsy.

### 4.2. Patient and Post-Mortem Tumor Site Selection

Eligible patients were identified through reviewing appointment lists, recommendations from tumor conferences, and direct contact with the treating physicians. Complete autopsies were performed according to standard procedures. Tissue samples were obtained from all macroscopically apparent and clinically known tumor sites. Tissue material was formalin-fixed and paraffin-embedded following the standard procedure of the Institute of Pathology and Molecular Diagnostics at the University Hospital of Augsburg. Post-mortem tissue was stained with hematoxylin and eosin and evaluated by a pathologist. Samples were macro-dissected, targeting tumor content exceeding 80%. Eight tumor samples per patient were obtained for this study, including the primary tumor if available. Manifestations showing progression during ongoing therapy based on CT, MRI, or PET were prioritized.

### 4.3. DNA Isolation and Sequencing of Tumor Material

DNA was extracted from the annotated tumor areas following the AVENIO Tumor DNA Isolation and QC kit (Roche Holding AG, Basel, Switzerland) with the AVENIO Tumor Cleanup and Capture Beads (Roche). Libraries were prepared using the AVENIO Tumor Library Prep Kit (Roche) with the AVENIO Tumor Surveillance Panel V1 (Roche), which targets 197 cancer-related genes spanning approximately 198 kilobases (kb) of genomic regions. Libraries were sequenced using paired-end 150 bp sequencing (dual indexing), aiming at a coverage of approximately 20,000-fold on an Illumina Next-Seq 500/550 platform (Illumina, San Diego, CA, USA). Twenty-four barcoded samples were sequenced per flow cell.

### 4.4. Cell-Free DNA (cfDNA) Isolation and CAPP-Seq Based Analysis

Peripheral blood (PB) was collected in the context of ALPS and processed within two hours. Plasma was separated by two centrifugation steps at 2000× *g* for 10 min each. Plasma samples were stored at −80 °C in the ACBB for long-term storage.

cfDNA was extracted from four mL of plasma, and libraries were prepared using the AVENIO cfDNA Isolation kit and the AVENIO Library Prep Kit with the ctDNA Surveillance Kit V1 (Roche Holding AG, Basel, Switzerland), respectively. cfDNA concentrations were measured before library preparation using fluorometric methods (Qubit 1x dsDNA HS Assay Kit; ThermoFisher Scientific, Waltham, MA, USA). Quality of both the isolated cfDNA and the library preparation was assessed based on size distribution (Bioanalyzer High Sensitivity DNA assay; Agilent Technologies, Santa Clara, CA, USA). Libraries were sequenced using paired-end 150 bp sequencing, targeting approximately 20,000 folds on an Illumina Next Seq550 or on an NovaSeq 6000 platform (both Illumina, San Diego, CA, USA). Sequencing was performed by the NGS Competence Center Tübingen (NCCT) ensuring a minimum analytical sensitivity of <0.1%, compliant with the Rili-Baek guidelines stipulating a minimum sensitivity of 0.5% for DNA from cell free body liquids [53].

### 4.5. Bioinformatic and Statistical Analysis

Base Call Intensities (BCL) files were used as input for the proprietary pre-analysis pipeline (AVENIO Oncology Analysis Software, Version 2.1.0) as well as the input DNA Q-Ratio and input DNA Mass. The resulting unfiltered set of SNVs was additionally annotated with VEP (Version 110.1) [54].

Initially, NGS tissue results were filtered for germline variants from blood samples, when available, or manually filtered based on variant allele frequencies (VAFs) (Appendix A).

Subsequently, variants removed from TBx results were also excluded from LBx analyses. An additional upper VAF threshold of 0.35 was applied to LBx variants to minimize non-somatic mutation inclusion. Potential artifacts were identified by analyzing the alternate allele depth (ALTDP), and values below 11 were discarded (Appendix A). Each annotated alteration was categorized as synonymous or missense.

Statistical analyses and data visualization utilized Python (3.11) and R (4.3.2). Descriptive statistics, hierarchical clustering analyses, and regressions were performed with pandas and scikit-learn packages. Data visualization was performed with matplotlib and seaborn packages. Oncoprint visualizations were generated in R using ComplexHeatmap in combination with ggplot2. Comparisons between groups were performed using the Mann–Whitney U test, with *p*-values < 0.05 considered statistically significant.

## 5. Conclusions

In conclusion, this study underscores the complementary roles of LBx and TBx in capturing the full spectrum of the tumor mutational landscape. Integrating LBx into clinical workflows alongside TBx may enhance the understanding of tumor genetics and individual mutational trajectories.

Future research should assess the clinical significance of LBx-exclusive mutations and further investigate the role of integrating LBx into routine molecular pathology workflows.

## Figures and Tables

**Figure 1 ijms-26-07614-f001:**
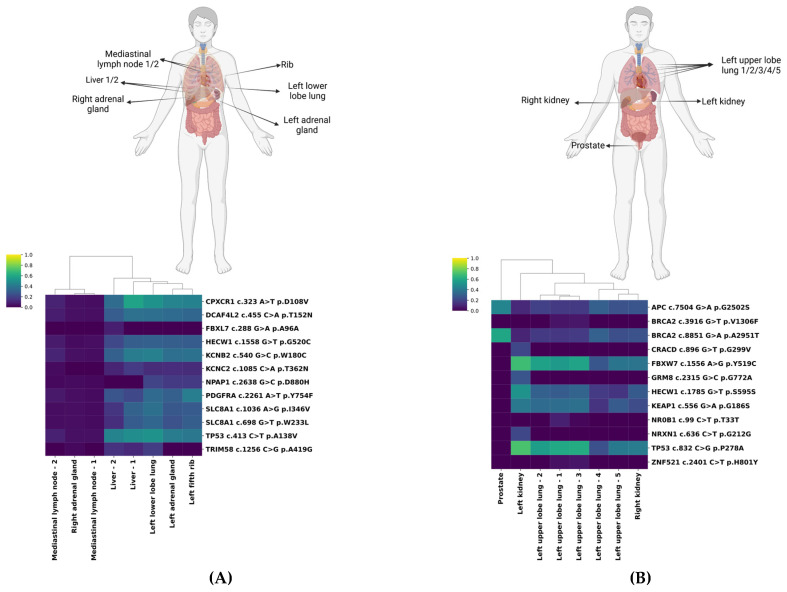
Exemplary hierarchical clustering of biopsied tumor lesions. Homunculi and heatmaps of Patient 4 (**A**) and Patient 5 (**B**) with hierarchical clustering visualizing the similarity in the mutational pattern of the lesions, including the corresponding variant allele frequencies (VAFs). The heatmap encodes tissue VAFs ranging from 0.0 (dark blue) to 1.0 (yellow).

**Figure 2 ijms-26-07614-f002:**
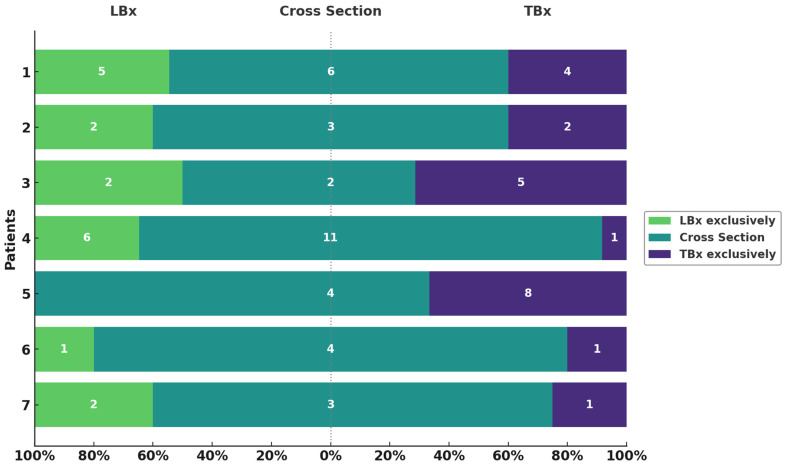
Variants exclusively detected in LBx or TBx and their overlap. Overview of variants per patient (y-axis) found in LBx exclusively (green), TBx exclusively (violet), and the intersection (turquoise). Respective numbers of variants are depicted in each bar. Proportions are shown on the x-axis as percentages.

**Figure 3 ijms-26-07614-f003:**
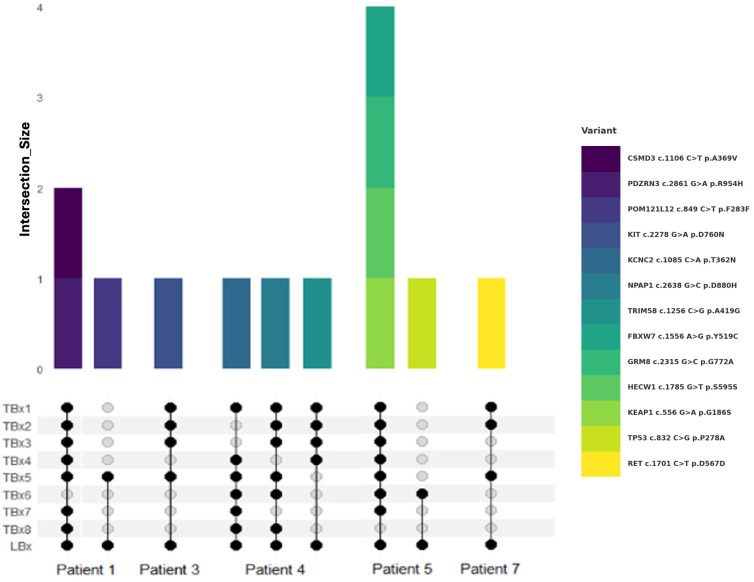
Alterations detected by LBx but not all TBx sites. Venn diagram visualizes the intersection between TBx lesions and LBx (bottom) for different patients. Lesion sites are summarized as TBx1-8. Colored bars encode corresponding variants shown on the right-hand side (top).

**Figure 4 ijms-26-07614-f004:**
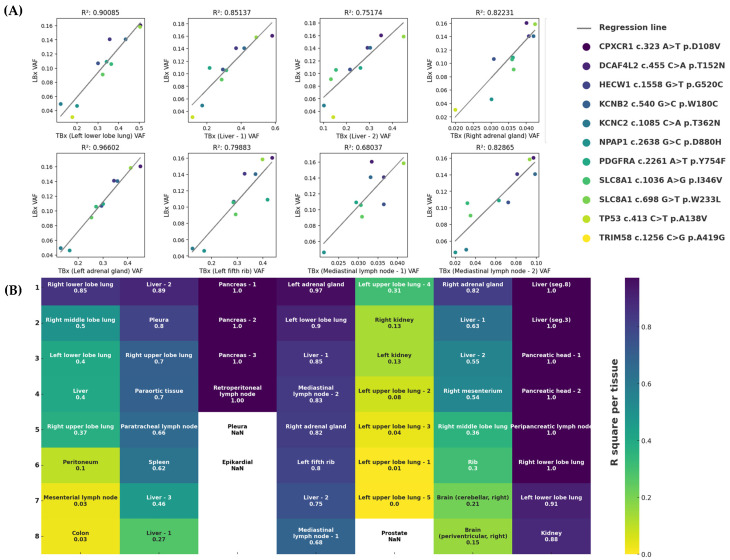
Correlation analysis between TBx and LBx with corresponding R^2^ values. (**A**) Exemplary linear regressions between VAFs of mutations in the intersection of Patient 4. Analyses were conducted for each lesion individually. Variants are described in the legend. Corresponding R^2^ values are shown above each subfigure. (**B**) Summary of R^2^ values in descending order for each lesion patient-wise (upper x-axis). R^2^ values range from 0.0 (yellow) to 1.0 (blue). No analysis is available for lesions with fewer than two alterations at the intersection (NaN: Not a Number).

## Data Availability

The data that support the findings of this study are available from the corresponding author upon reasonable request.

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
