# Peer review of "Dissecting Tumor Heterogeneity by Liquid Biopsy—A Comparative Analysis of Post-Mortem Tissue and Pre-Mortem Liquid Biopsies in Solid Neoplasias"

_ijms, 2025, doi:10.3390/ijms26157614_

Round 1

Reviewer 1 Report

Comments and Suggestions for Authors

The manuscript «Dissecting tumor heterogeneity by liquid biopsy - a comparative analysis of post-mortem tissue and pre-mortem liquid biopsies in solid neoplasias» describes results of a study dedicated to the assessment of the concordance of TBx (across various tumor sites) and LBx findings (regarding mutational profiling). I congratulate the Authors on successfully conducting a study with such an intricate design. The methodology is robust regarding all aspects of the study (from sampling to bioinformatics). The Authors did an outstanding job visualizing their findings in figures and tables, so the data was quite easy to interpret. The supplementary materials contain all necessary information. The limitations of the study are presented as well. The absence of a 100% concordance between TBx and LBx findings was known prior to this study, but it does not mean that it lacks novelty. To my mind, this study is of great interest to readers and of high value overall, as it provides exact metrics for the heterogeneity observed. Therefore, its results might be useful for both clinical and scientific (regarding study designs, data interpretation, etc.) decision-making.

However, there are several minor concerns / questions which I have.

  1. In general, the manuscript is written in good English, although it still requires minor correction of grammar and typos.
  2. It seems that the abstract is larger than it should be according to the instructions for authors section of the web site of the journal.
  3. Please ensure that all abbreviations are explained in the text and/or in the specific section. For example, “DP”.
  4. Figure 4B. I think it is better to move the caption “R square per tissue” to the right side of the graph where the actual values are.
  5. The authors mentioned clonal hematopoiesis which might explain the detection of LBx-exclusive mutations. Is it possible to examine the mutations listed in Supplementary Table 1 and compare them to the list of known clonal hematopoiesis-related mutations? Are these mutations relevant in terms of drug resistance?

Author Response

Dear Dr. Fernandez, Dear Dr. Paolillo,

Dear reviewers,

We would like to take this opportunity to express our sincere gratitude for evaluating our manuscript and considering it for publication in in your Special Issue “Liquid Biopsies in Oncology” in the International Journal of Molecular Sciences.

Furthermore, we specifically thank the reviewers for their time and effort spent critically reviewing our manuscript and providing us with constructive feedback that improves the quality of our manuscript.

Please find a point-by-point response below. We have addressed all technical issues and refined the figure quality. Main figure legends are now provided in the manuscript while main figures were separately attached.

We have marked all changes in the revised manuscript.

In summary, we believe that the revised manuscript has been improved. It is a privilege to contribute to the scientific community through our revised manuscript in your journal.

Yours sincerely,

For the authors:

Maximilian Schmutz, MD

Rainer Claus, MD

Reviewer 1

The manuscript «Dissecting tumor heterogeneity by liquid biopsy - a comparative analysis of post-mortem tissue and pre-mortem liquid biopsies in solid neoplasias» describes results of a study dedicated to the assessment of the concordance of TBx (across various tumor sites) and LBx findings (regarding mutational profiling). I congratulate the Authors on successfully conducting a study with such an intricate design. The methodology is robust regarding all aspects of the study (from sampling to bioinformatics). The Authors did an outstanding job visualizing their findings in figures and tables, so the data was quite easy to interpret. The supplementary materials contain all necessary information. The limitations of the study are presented as well. The absence of a 100% concordance between TBx and LBx findings was known prior to this study, but it does not mean that it lacks novelty. To my mind, this study is of great interest to readers and of high value overall, as it provides exact metrics for the heterogeneity observed. Therefore, its results might be useful for both clinical and scientific (regarding study designs, data interpretation, etc.) decision-making.

However, there are several minor concerns / questions which I have.

  1. In general, the manuscript is written in good English, although it still requires minor correction of grammar and typos.

We sincerely thank the reviewer for their thorough and insightful evaluation of our manuscript "Dissecting tumor heterogeneity by liquid biopsy – a comparative analysis of post-mortem tissue and pre-mortem liquid biopsies in solid neoplasias." We greatly appreciate your recognition of the complexity of the study design, the robustness of the methodology, and the clarity of our data visualization and supplementary materials. Your constructive feedback is valuable and helps reinforce the significance and potential impact of our findings.

In response to your suggestions, we have carefully revised the manuscript to address grammar and typographical errors, ensured consistency in abbreviations, updated the caption of Figure 4B, and rephrased several smaller sections to improve clarity and English wording.

  1. It seems that the abstract is larger than it should be according to the instructions for authors section of the web site of the journal.

We have adjusted the length of the abstract from 297 words to 185 words in single paragraph format. Please find the updated abstract below:

“Tumor heterogeneity encompasses genetic, epigenetic, and phenotypic diversity, impacting treatment response and resistance. Spatial heterogeneity occurs both inter- and intra-lesionally, while temporal heterogeneity results from clonal evolution. High-throughput technologies like next-generation sequencing (NGS) enhance tumor characterization, though traditional biopsies insufficiently capture heterogeneity. Liquid biopsy (LBx), analyzing circulating tumor DNA (ctDNA), provides a minimally invasive alternative, offering real-time tumor evolution insights and identifying resistance mutations overlooked by tissue biopsies. This study evaluates LBx’s capability to capture tumor heterogeneity by comparing genetic profiles from multiple metastatic lesions and LBx samples. Eight patients from the Augsburger Longitudinal Plasma Study with diverse cancers provided 56 postmortem tissue samples, compared against pre-mortem LBx-derived circulating-free DNA sequenced by NGS. Tissue analyses revealed significant mutational diversity (4–12 mutations per patient, VAFs: 1.5–71.4%), with distinct intra- and inter-lesional heterogeneity. LBx identified 51 variants (4–17 per patient, VAFs: 0.2–31.1%), with 33–92% overlap with tissue mutations. Notably, 22 tissue variants were absent in LBx, whereas 18 LBx-exclusive variants were detected (VAFs: 0.2–2.8%). LBx effectively captures tumor heterogeneity but should be utilized complementary to tissue biopsies for comprehensive genetic profiling.”

  1. Please ensure that all abbreviations are explained in the text and/or in the specific section. For example, “DP”.

Please see answer 1)

  1. Figure 4B. I think it is better to move the caption “R square per tissue” to the right side of the graph where the actual values are.

Please see answer 1)

  1. The authors mentioned clonal hematopoiesis which might explain the detection of LBx-exclusive mutations. Is it possible to examine the mutations listed in Supplementary Table 1 and compare them to the list of known clonal hematopoiesis-related mutations? Are these mutations relevant in terms of drug resistance?

We thank the reviewer for raising the important issue of clonal hematopoiesis of indeterminate potential (CHIP), which is increasingly recognized as a confounding factor in the interpretation of liquid biopsy results. As clonal expansion driven by somatic mutations is common in the aging hematopoietic system, distinguishing tumor-derived mutations from CHIP-related alterations can be essential to avoid misinterpretation.

To address this concern, we reviewed all mutations listed in Supplementary Table 1 and compared them to a curated list of genes recurrently associated with CHIP, including:

DNMT3A, TET2, ASXL1, JAK2, TP53, SF3B1, PPM1D, SRSF2, IDH1, IDH2, U2AF1, KRAS, NRAS, CTCF, CBL, GNB1, BRCC3, PTPN11, GNAS, BCOR, BCORL1

and

BRAF, CALR, CEBPA, CRBBP, CSF1R, CSF3R, CUX1, ETV6, EZH2, GATA2, JAK3, KDM6A, KIT, KMT2A, MPL, MYD88, NOTCH1, PHF6, PIGA, PRPF40B, PTEN, RAD21, RUNX1, SETBP1, SF1, SF3A1, SMC1A, SMC3, STAG2, STAT3, U2AF2, WT1, ZRSR2

We found only one overlapping gene, KIT, with a variant classified as a variant of unknown significance (VUS). Based on current knowledge, we consider this variant unlikely to impact the interpretation of our findings, particularly regarding clinically relevant resistance mechanisms.

We have added a corresponding paragraph to the discussion section of the revised manuscript (p11, line 840) to address this point. Additionally, we acknowledge that future studies would benefit from matched sequencing of plasma and leukocytes to more robustly differentiate between tumor-derived and hematopoietic-origin variants.

“In response to the potential confounding role of clonal hematopoiesis of indeterminate potential (CHIP) in interpreting LBx-exclusive mutations, we systematically evaluated all alterations listed in Supplementary Table 1 against a curated set of genes frequently associated with CHIP. This list includes canonical drivers such as DNMT3A, TET2, ASXL1, JAK2, TP53, SF3B1, PPM1D, SRSF2, IDH1, IDH2, and U2AF1, among others commonly implicated in age-related clonal expansions of hematopoietic lineages. Among the variants identified, only one—located in KIT—overlapped with this CHIP-associated gene set. However, this variant was classified as a variant of unknown significance (VUS), and thus is unlikely to have clinical relevance in the present context, particularly regarding drug resistance. Based on current evidence, we consider it unlikely that CHIP-associated mutations significantly influenced our LBx findings. Nonetheless, we emphasize the importance of incorporating paired plasma and leukocyte sequencing in future studies to more definitively distinguish tumor-derived from hematopoietic-origin variants.”

We have moreover (p.7 line 569) added a sentence into our results section (“Liquid and tissue biopsies reveal partially overlapping mutation profiles”).

“Among the variants identified, one, located in KIT, overlapped with genes associated with clonal hematopoesis of indeterminate potential (CHIP) and can therefore not be confidently associated as of tumor origin.”

Reviewer 2 Report

Comments and Suggestions for Authors

In their manuscript, Mogele and colleagues presented a detailed study comparing liquid biopsy with traditional tissue biopsy. Samples were taken from patients with different types of tumors and sites of metastasis and the effectiveness of the two techniques in detecting mutations and tumor heterogeneity was compared. The manuscript is very well written, with a well-founded introduction that helps to understand the experimental findings. The results obtained are relevant and indicate the importance of the complementarity of the two techniques. I believe that the specialized literature in the field will benefit greatly from the publication of this manuscript. My only recommendation would be to improve the quality and resolution of the figures presented; the texts contained in them are blurred and difficult to read.

Author Response

Dear Dr. Fernandez, Dear Dr. Paolillo,

Dear reviewers,

We would like to take this opportunity to express our sincere gratitude for evaluating our manuscript and considering it for publication in in your Special Issue “Liquid Biopsies in Oncology” in the International Journal of Molecular Sciences.

Furthermore, we specifically thank the reviewers for their time and effort spent critically reviewing our manuscript and providing us with constructive feedback that improves the quality of our manuscript.

Please find a point-by-point response below. We have addressed all technical issues and refined the figure quality. Main figure legends are now provided in the manuscript while main figures were separately attached.

We have marked all changes in the revised manuscript.

In summary, we believe that the revised manuscript has been improved. It is a privilege to contribute to the scientific community through our revised manuscript in your journal.

Yours sincerely,

For the authors:

Maximilian Schmutz, MD

Rainer Claus, MD

Reviewer 2

In their manuscript, Mogele and colleagues presented a detailed study comparing liquid biopsy with traditional tissue biopsy. Samples were taken from patients with different types of tumors and sites of metastasis and the effectiveness of the two techniques in detecting mutations and tumor heterogeneity was compared. The manuscript is very well written, with a well-founded introduction that helps to understand the experimental findings. The results obtained are relevant and indicate the importance of the complementarity of the two techniques. I believe that the specialized literature in the field will benefit greatly from the publication of this manuscript. My only recommendation would be to improve the quality and resolution of the figures presented; the texts contained in them are blurred and difficult to read.

We thank the reviewer for their positive assessment of our manuscript and their thoughtful remarks. We appreciate your recognition of the clarity of the writing, the relevance of the findings, and the complementary value of liquid and tissue biopsy approaches in assessing tumor heterogeneity. We are especially grateful for your suggestion regarding figure quality and have updated all figures to ensure improved readability in the final submission.